# Intravenous Iron Therapy to Treat Anemia in Oncology: A Mapping Review of Randomized Controlled Trials

**Jayne Lim [1], Michael Auerbach [2], Beth MacLean [1], Annas Al-Sharea [1] and Toby Richards [1,*]**

1    School of Medicine, University of Western Australia, Perth, WA 6009, Australia
2    Department of Medicine, Georgetown University, Washington, DC 20007, USA
*    Correspondence: toby.richards@uwa.edu.au

**Abstract:** Anemia is a common problem when patients present with cancer, and it can worsen during treatment. Anemia can directly impact the cognitive and physical quality of life and may impair fitness for oncological therapy. The most common cause of anemia is iron deficiency. Newer intravenous (IV) iron formulations offer a safe and rapidly effective treatment option. We performed a systematic mapping review of randomized controlled trials (RCTs) evaluating intravenous iron therapy in patients with cancer and anemia and their outcomes. A total of 23 RCTs were identified. The median number of patients enrolled was 104 (IQR: 60–134). A total of 5 were focused on surgical outcomes (4 preoperative, 1 postoperative), and 15 were in adjuvant therapies for a variety of tumor types (breast, colorectal, lung, gynecological, myeloid, and lymphomas), 10 of which were in combination with erythropoietin-stimulating agents (ESAs) therapy, 2 in radiotherapy, and 1 in palliative care. Overall, the studies reported that the use of IV iron increased hemoglobin concentration and decreased transfusion rates during different cancer treatment regimes. IV iron can be administered safely throughout the cancer treatment pathway from primary surgery to the palliative setting. More studies are needed to demonstrate net clinical outcomes.

**Keywords:** anemia; oncology; mapping; randomized controlled trials

## 1. Introduction

Anemia is a common problem in cancer patients; one-third have anemia at diagnosis, and half develop anemia during chemotherapy and/or radiation therapy [1]. Anemia is defined by the World Health Organization as a hemoglobin concentration (Hb) < 120 g/L in women and <130 g/L in men [1]. Anemia is caused by one or more primary mechanisms: blood loss, hemolysis, and reduction in erythropoiesis [2]. Blood loss is a direct result of gastrointestinal tumors and bone marrow replacement by cancer cells. Bone marrow suppression, cancer-related cachexia, and hepcidin-mediated iron sequestration are indirect effects of inflammation [2]. Anemia is associated with fatigue, impaired physical function, and decreased quality of life [3]. The severity of anemia is associated with a decreased quality of life [4]. The impact of anemia on "fitness" has a direct impact on oncological therapy, where performance status impacts fitness for treatment [5]. This is particularly relevant prior to surgery when either neoadjuvant or adjuvant therapy is indicated [6].

The problem of anemia has been highlighted in surgical patients, where one in three has preoperative anemia [7]. Preoperative anemia itself is independently associated with increased morbidity and mortality, as well as the risk of red blood cell (RBC) transfusion, postoperative complications, and increased hospital stay [8]. Blood loss during operation and hospital stay results in most patients being discharged with anemia, one in four with Hb < 100 g/L, which is also associated with increased rates of readmission for complications and delayed recovery [9]. After major surgery for solid tumors, which often require adjuvant therapy, a delay in initiation is associated with inferior overall treatment and survival outcomes in several meta-analyses [10,11].

Current treatment options for anemia include the use of blood transfusions, erythroid-stimulating agents (ESAs), or iron [12,13]. Although the use of blood transfusions can rapidly increase Hb, there is caution as database analyses suggest that it may be associated with adverse outcomes and potentially an immunomodulated risk of increased cancer recurrence. ESAs are effective in increasing Hb, and guidelines limit administration to Hb levels less than 100 g/L due to some safety concerns [14–17]. Prospective studies and meta-analyses investigating ESAs that have followed did not demonstrate negative effects [18–20]. Iron therapy has traditionally focused on oral iron. However, side effects in the gastrointestinal tract often mean that only a minority of cancer patients, who may already experience nausea, diarrhea, or constipation as a side effect of their cancer therapy, can tolerate oral iron. Intravenous (IV) iron has had limited use due to historical concerns about anaphylaxis (high molecular weight iron dextran products are no longer available for clinical use) or low doses (200 mg of iron sucrose) [21]. However, the last twenty years have seen the development of new IV iron carbohydrate preparations that allow a large dose (1000 mg) to be administered safely in 15 min [21]. The use of IV iron has increased exponentially in many areas of clinical practice, including surgery, women's health, cardiology, and hematology. IV iron can bypass the hepcidin-mediated iron sequestration effect of inflammation in cancer. Therefore, we wish to review the evidence for IV iron in oncology to identify, describe, and map the RCTs that evaluate IV iron therapy in cancer patients with anemia and their results.

## 2. Materials and Methods

A mapping review was performed to map out and categorize the existing literature [22]. This review was conducted and reported according to the Preferred Reporting Items for Systematic Review and Meta-analysis (PRISMA) [23] as part of a broader protocol registered with PROSPERO (CRD42019148956).

### 2.1. Eligibility Criteria

The participants, interventions, comparators, outcomes, and study types (PICOS) framework was used to guide the eligibility criteria.
Inclusion criteria:

- Population: Adult patients with anemia as defined by the WHO criteria: Hb < 130 g/L for men and Hb < 120 g/L for women and cancer;
- Intervention: IV iron, regardless of dose or frequency;
- Comparators: IV iron, oral iron, placebo, standard of care, or no treatment;
- Outcomes: Studies evaluating the risk of receiving red cell transfusion, hematological measures, and quality of life;
- Study type: randomized controlled trials.

### 2.2. Search Strategy

We included studies from 1966 to 1 March 2023, which were included in the previous reviews, and conducted another search for studies published from 1 June 2019 to 1 March 2023 [24,25]. We searched: CENTRAL (The Cochrane Central Register of Controlled Trials), MEDLINE (Ovid), EMBASE (Ovid), and CINAHL (Cumulative Index to Nursing and Allied Health Literature). We also searched ISI Web of Science: Science Citation Index Expanded (SCI-EXPANDED) and ISI Web of Science: Conference Proceedings Citation Index-Science (CPCI-S). The searches were not restricted by blinding, language, or publication status. We screened the reference lists of included studies for further eligible studies. The key terms included iron, ferrous, ferric, and an(a)emia/c. Full details of the search strategy are available (Appendix A).

### 2.3. Study Selection and Data Extraction

Two independent reviewers (JL and BM) performed the initial selection of titles and abstracts for all articles using the Rayyan Web application. Two reviewers (JL and

BM) performed a full-text assessment to identify the trials for inclusion independently of each other, listing the excluded studies and the reason for the exclusion using EndNote (Version 20). A reviewer (JL) extracted and tabulated the data from the included studies for this review. The following data were extracted: year of publication, study design, indication, number of patients, interventions, comparisons, and outcomes.

*2.4. Data Synthesis*

The results are presented in a summary table and describe the characteristics of the included study. A narrative synthesis of the included studies was performed to map IV iron therapy along the cancer treatment journey (Figure 1).

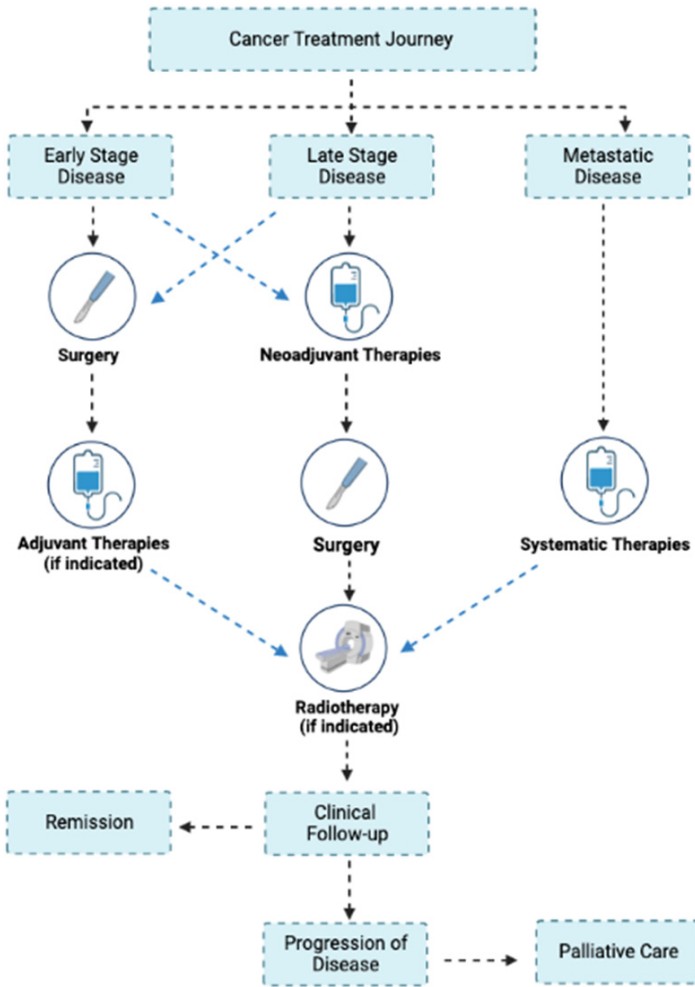

**Figure 1.** A general overview of the cancer treatment journey. Created with BioRender.com accessed on 8 May 2023.

## 3. Results

A total of 8493 records were retrieved from the search performed on 7 March 2023, and 138 full-text articles were evaluated for eligibility Figure 2 [26]. A total of 23 studies met the inclusion criteria for this review. The median number enrolled was 104 (IQR: 60–134). A summary of the included studies is presented in Table 1.

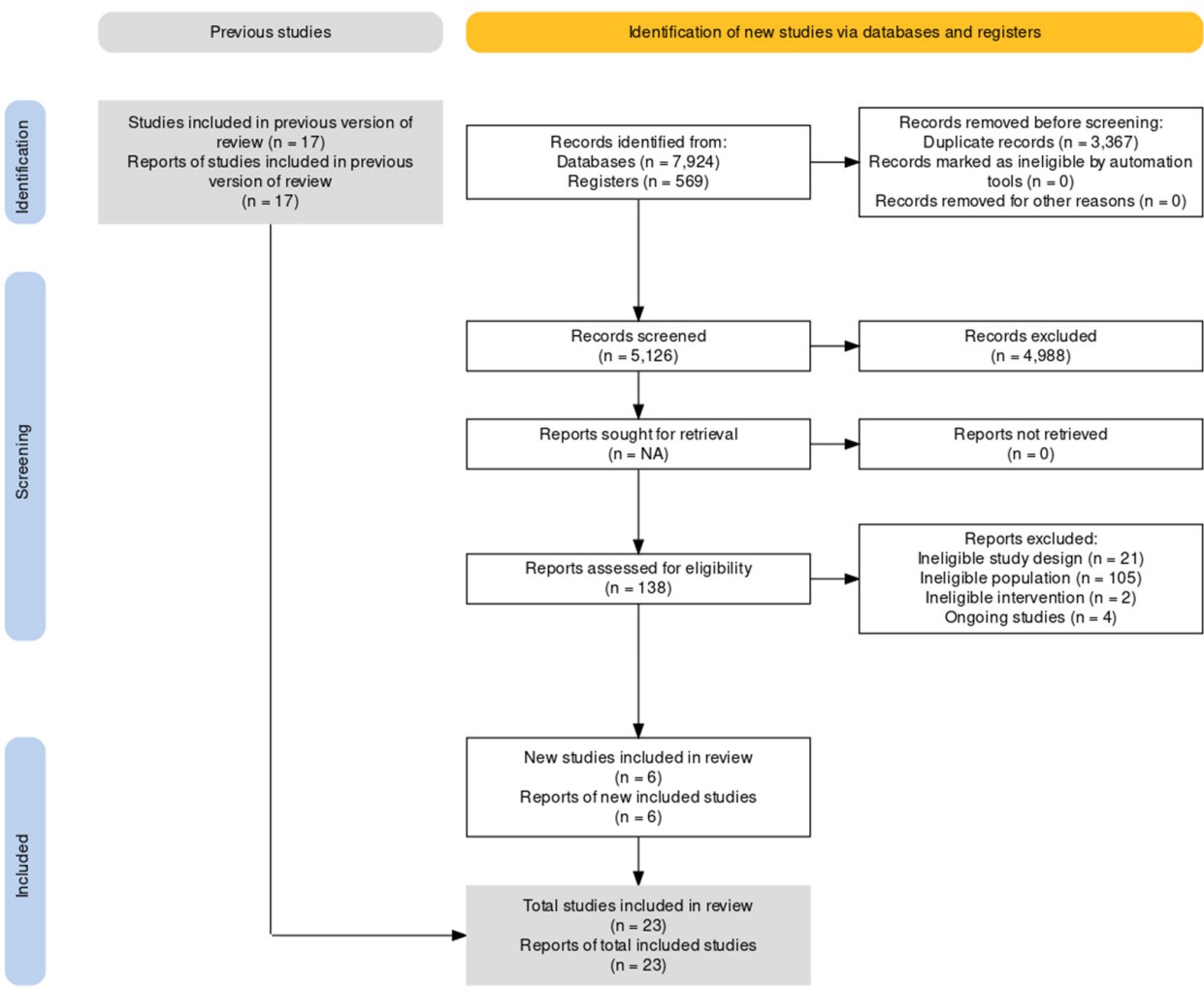

**Figure 2.** Preferred reporting items for systematic reviews flowchart of the study selection process.

**Table 1.** Summary of characteristics of included trials.

| Author | Study Type | Indication | Inclusion Criteria | N | Treatment | Results |
|---|---|---|---|---|---|---|
| Ansari Nejad 2016 [27] | Prospective, single-center, open-label RCT | Stage 3/4 colon cancer undergoing chemotherapy | Hb ≤ 120 g/L for women and ≤130 g/L for men, SF < 30 µg/L | 60 | G1: Oral ferrous sulfate 65 mg t.i.d for 8 wk<br>G2: IV FCM (1500 mg for patients weighting < 70 kg or 2000 mg for patients weighting > 70 kg) | Significantly higher Hb in IV FCM group (138.6 ± 7.4 g/L vs. 116.7 ± 12.8 g/L) |
| Anthony 2011 [28] | Prospective, multicentre, open-label RCT | Cancer and/or chemotherapy-induced anemia | Hb ≤ 100 g/L | 375 | G1: IV IS (7 mg/kg up to 500 mg) × 3 times per wk with 1 to 3 week intervals + ESAs<br>G2: No iron + ESAs | Higher Hb in IV IS + ESAs Improved FACIT fatigue scores in IV IS group |
| Auerbach 2004 [29] | Prospective, multicentre, open-label RCT | Cancer and/or chemotherapy-induced anemia | Hb ≤ 105 g/L, SF ≤ 200 µg/L or SF ≤ 300 µg/L with TSAT ≤ 19% | 157 | Epoetin alfa 40,000 U/wk in addition to:<br>G1: No iron<br>G2: Oral ferrous sulfate 325 mg b.i.d<br>G3: IV bolus iron dextran repeated 100 mg<br>G4: IV iron dextran total dose infusion | Greater mean Hb increase in both IV iron groups compared to oral iron and no iron groups Hb response in IV groups compared to oral iron and no iron groups (68% *v* 25%) QoL improvement in IV iron groups |
| Auerbach 2010 [30] | Prospective, multicentre, double-blind, 2 × 2 factorial RCT | Cancer and/or chemotherapy in active nonmyeloid malignancies | Hb ≤ 100 g/L Exclusion: TSAT < 15% and SF < 10 µg/L | 243 | G1: SC DA 500 µg Q3W + no iron<br>G2: SC DA 500 µg Q3W + IV iron dextran 400 µg Q3W<br>G3: SC DA 300 µg Q3W + no iron<br>G4: SC DA 300 µg Q3W + IV iron dextran 400 µg Q3W | Higher proportion achieved Hb ≥ 110 g/L in IV iron groups (82% *v* 72%). Clinically significant increase in FACT-F scores were 67%, 100%, 65%, and 63%, respectively |
| Bastit 2008 [31] | Prospective, multicentre, open-label RCT | Nonmyeloid malignancy undergoing chemotherapy | Hb ≤ 110 g/L Exclusion: TSAT < 15% and SF < 10 µg/L), SF > 800 µg/L | 396 | G1: IV FG or IS 200 mg Q3W as single dose or two doses + SC DA 500 µg Q3W<br>G2: Oral iron or no iron + SC DA 500 µg Q3W | Higher proportion achieved Hb target in IV iron group (86% *v* 73%). Lower transfusion rate in IV iron group from week 4 to end of trial period (9% *v* 20%) No differences in QoL scores |
| Dangsuwan 2010 [32] | Prospective, single-center, open-label RCT | Gynecological cancer receiving first-line chemotherapy after primary surgery | Hb < 100 g/L | 44 | G1: IV IS 200 mg, single dose<br>G2: Oral ferrous sulfate 200 mg t.i.d | Lower RBC transfusion rate in IV IS (22.7% *v* 63.6%). Higher mean Hb (100 ± 8 g/L *v* 95 ± 9 g/L) No difference in change in QoL scores |

**Table 1.** *Cont.*

| Author | Study Type | Indication | Inclusion Criteria | N | Treatment | Results |
|---|---|---|---|---|---|---|
| Dickson 2023 [33] | Prospective, multicentre, placebo-controlled feasibility RCT | Advanced solid tumors | Hb < 120 g/L for women and <130 g/L men | 34 | G1: IV FDI 20 mg/kg/week<br>G2: Placebo (250 mL 0.9% sodium chloride) | Feasible trial according to recruitment and attrition rates. Trial was not powered to detect a significant difference in Q5D5L, QLQ-C30, and the FACIT-F scores |
| Dreyer 2017 [34] | Prospective, multicentre, open-label RCT | Locally advanced cervical cancer requiring primary radiation treatment | Hb ≤ 120 g/L | 43 | Limited RBC transfusion to Hb = 60 g/L + IV IS (Ganzoni formula)—most patients 1 g total dose.<br>RBC transfusion to Hb > 120 g/L<br>Both groups received oral ferrous fumarate 400 mg on discharge. | A steady rise of Hb in the IV IS group to week 12.<br>Transfusion group showed a steady decline of about 5 g/L per week post-treatment |
| Edwards 2009 [35] | Prospective, single-center, placebo-controlled RCT | Elective surgery for suspected colorectal cancer | Hb ≤ 125 g/L for women, Hb ≤ 135 g/L for men | 60 | G1: IV IS 600 mg in two divided doses, at least 24 h apart, 14 days preoperatively<br>G2: placebo in two divided doses, at least 24 h apart, 14 days preoperatively | No difference in Hb or transfusion rates |
| Fung, 2022 [36] | Prospective, single-centre, open-label pilot RCT | Elective colorectal cancer surgery | Hb < 130 g/L, SF < 30 µg/L or SF = 30–100 µg/L with TSAT < 20% | 40 | G1: IV IIM 20 mg/kg (up to 1000 mg) preoperatively<br>G2: Usual preoperative care (no iron) | Higher mean Hb change before surgery in IV IIM (7.8 g/L *v* 1.7 g/L)<br>No differences in QoR-15 and DAH30 at POD 30 |
| Hajigholami 2021 [37] | Prospective Single-centre open-label RCT | Metastatic and non-metastatic carcinoma undergoing chemotherapy | Hb ≤ 120 g/L<br>Exclusion:<br>SF > 500 µg/L | 79 | G1: IV IS 100 mg at each chemotherapy session + SC EPO 150 units/kg SC three times a week)<br>G2: Oral ferrous sulfate 100 mg t.i.d for six wks + EPO 150 units/kg SC three times a week) | No significant between-group differences in Hb increase (114 ± 16 *v* 112 ± 14 g/L). Physical index score increased in IV group. No significant between-group differences in QLQ-C30 scores |
| Hedenus 2007 [38] | Prospective, multicentre, open-label RCT | Lymphoproliferative malignancy not requiring chemotherapy or blood transfusions | Hb 90–110 g/L<br>Exclusion:<br>SF > 800 µg/L | 60 | G1: IV IS 100 mg/wk for 6 wks followed by 100 mg Q2W for 8 wks + SC EPO 30,000 IU/wk for 16 weeks<br>G2: SC EPO 30,000 IU/wk for 16 weeks | Higher Hb increase in IV iron group (93% *v* 53%) |

**Table 1.** *Cont.*

| Author | Study Type | Indication | Inclusion Criteria | N | Treatment | Results |
|---|---|---|---|---|---|---|
| Hedenus 2014 [39] | Prospective, multicentre, open-label RCT | Indolent lymphoid malignancy with cancer-related anemia | Hb 85–105 g/L and SF > 30 µg/L for women or >40 µg/L for men, TSAT ≤ 20% | 17 | G1: IV FCM 1 g, (>50 kg single dose, 500 mg two weeks apart if <50 kg) G2: Control (no treatment, symptomatic management according to local practice) | Significantly higher mean change in Hb in IV FCM group at 8 weeks (Hb = 21 g/L vs. 11 g/L) |
| Henry 2007 [40] | Prospective, multicentre, open-label RCT | Patients with chemotherapy anemia | Hb ≤ 110 g/L SF ≥ 100 µg/L TSAT ≥ 15% | 187 | G1: IV FG 125 mg/wk for 8 wks + SC EPO 40,000 U/wk for 12 wks G2: Oral ferrous sulfate 325 mg t.i.d SC EPO 40,000 U/wk for 12 wks G3: No iron + SC EPO 40,000 U/wk for 12 wks | Hb response was 73% for FG, 46% for oral iron, and 41% for no iron |
| Keeler 2017 [41] | Prospective, multicentre, open-label RCT | Elective colorectal cancer surgery | Hb ≤ 110 g/L for women and ≤120 g/L for men | 101 | G1: IV FCM 1–2 g (up to two doses with one week apart) preoperatively G2: Oral ferrous sulfate 200 mg b.i.d. at least two weeks before surgery | No difference in transfusion rates Hb increase in IV FCM (median 1.55 (i.q.r. 0.93–2.58) *v* 0.50 (−0.13 to 1.33) g/dl |
| Kim 2007 [42] | Prospective, single-centre, open-label RCT | Cervical cancer undergoing chemoradiotherapy | Hb ≤ 120 g/L | 75 | G1: IV IS 200 mg single infusion G2: Control (no iron) | Decreased transfusion requirement in IV IS group (40% *v* 64%) and mean transfusion units (1.87 *v* 3.58) |
| Laso-Morales 2022 [43] | Prospective single-centre, open-label RCT | Elective colorectal cancer surgery | Hb < 110 g/L after surgery | 104 | G1: IV FCM 1 g, single dose on POD1 G2: IV IS 200 mg (every 48 hrs from POD1 to discharge or up to the total dose equivalent using Ganzoni Formula | No differences in Hb, transfusion rates or length of stay Infection rate lower in IV FC (9.8% *v* 37.2) |
| Maccio 2010 [44] | Prospective, multicentre, open-label RCT | Advanced solid tumor undergoing chemotherapy | Hb ≤ 100 g/L SF ≥ 100 µg/L and ≤800 mg/dL and/or TSAT > 15% | 148 | G1: IV FG 125 mg weekly + EPO 30,000 UI SC weekly for 12 weeks G2: Oral lactoferrin 100 mg b.i.d + SC EPO 30,000 UI/wk for 12 weeks | No difference in Hb +1.6 g/dL V +1.8 g/dL for lactoferrin |
| Makharadze 2021 [45] | Prospective, multicentre, placebo-controlled RCT | Nonmyeloid malignancy undergoing chemotherapy | Hb: 80–110 g/L, SF:100–800 µg/L TSAT ≤ 35% | 244 | G1: IV FCM 15 mg/kg (single and total doses of 750 mg and ≤1500 mg, respectively) 7 days apart G2: Placebo normal saline 0.9% ≤250 mL of normal saline two infusions 7 days apart | Higher maintained Hb within 0.5 g/dL of baseline in IV FCM group (50.8% *v* 35.3%) |

**Table 1.** *Cont.*

| Author | Study Type | Indication | Inclusion Criteria | N | Treatment | Results |
|---|---|---|---|---|---|---|
| Noronha 2018 [46] | Prospective single-center, open-label RCT | Malignancy requiring chemotherapy | Hb < 120 g/L<br>SF < 100 µg/L<br>TSAT < 20%<br>or hypochromic RBC > 10% | 148 | G1: IV IS (Ganzoni Formula)<br>G2: Oral ferrous sulfate 100 mg t.i.d, started with cycle one of chemotherapy and continued until the end of cycle 2 | No difference between groups in change In Hb at 6 weeks (0.11 g/dL *v* 0.16 g/dL)<br>No difference in QoL |
| Pedrazzoli 2008 [47] | Prospective multicentre, open-label RCT | Breast, colorectal, lung, or gynaecologic cancer undergoing chemotherapy | Hb ≤ 120 g/L<br>SF ≥ 100 µg/L<br>TSAT ≥ 20% | 149 | G1: IV FG 125 mg/wk for first 6 wks + SC DA 150 µg/wk for 12 wks<br>G2: No iron + SC DA 150 µg/wk for 12 wks | Higher Hb response in IV FG/DA group (76.7% *v* 61.8%)<br>Faster Hb response from week 5 in IV FG/DA group |
| Steensma 2010 [48] | Prospective multicentre, open-label RCT | Nonmyeloid malignancy undergoing chemotherapy | Hb ≤ 110 g/L | 502 | SC DA 500 µg Q3W in addition to:<br>G1: IV FG 187.5 mg Q3W for 5 doses<br>G2: Oral ferrous sulfate 325 mg for 16 wks<br>G2: Oral placebo for 16 wks | No difference In Hb, transfusion, or QoL on intention-to-treat analysis |
| Talboom, 2023 [49] | Prospective, multicentre, open-label RCT | Elective colorectal cancer surgery | Hb < 120 g/L for women and <130 g/L for men<br>TSAT < 20% | 202 | G1: IV FCM 1–2 g (up to two doses with one wk apart) preoperatively<br>G2: Oral ferrous fumarate 200 mg t.i.d. until day before surgery | No difference In Hb normalization before surgery (17% *v* 16%), transfusion rates, length of stay, post-op complications. Higher Hb normalized in IV iron at 30 days post-op (60% *v* 21%). No difference in BFI and EQ5D scores. Improved scores in Role Functioning Scale in EORTC 30 and three symptom scales on the EORTC C29 in oral iron group |

Abbreviations: BFI, brief fatigue inventory; DAH30, days (alive and) at home within 30 days of surgery; DA, darbepoetin alfa; EPO, erythropoietin; FACIT, functional assessment of chronic illness therapy; FCM, ferric carboyxmaltose; FDI, ferric derisomaltose; FG, ferric gluconate; G, Group; IIM, iron isomaltoside; IS, iron sucrose; POD, postoperative day; QoL, quality of life score, QoR-15, 15-item quality of recovery; RBC, red blood cells; RCT, randomized controlled trial; SC, subcutaneous; SF, serum ferritin; TSAT, transferrin saturation.

### 3.1. Surgery

There were five RCTs performed on colorectal cancer patients undergoing elective surgery; four evaluated the efficacy of preoperative IV iron, and one compared different formulations of IV iron in the postoperative setting [35,36,41,43,49]. The studies included patients with anemia (defined as Hb <100 g/L–≤135 g/L), with two studies including iron parameters. In general, the most common primary endpoint was a change in Hb and a decrease in transfusion rates. IV iron increased Hb levels before and after surgery (7.8 g/L), but secondary endpoints of quality of life scores, length of hospital stay, postoperative complications, and adverse events showed limited differences between groups. In summary, the use of IV iron appears to correct laboratory values of Hb and may reduce the need for blood transfusion, but clinical trials have not shown direct patient benefit on clinical outcomes.

Detailed Description of the IVICA and FIT Trials

IVICA included 116 patients with anemia defined as 10 g/L below WHO definition, scheduled for elective colorectal cancer surgery a minimum of 14 days before surgery [41]. The intervention of 1000 mg to 2000 mg of IV ferric carboxymaltose (FCM) based on the patient's Hb and weight was compared to 200 mg of oral ferrous sulfate twice daily until the day of surgery. There was no difference in the primary endpoint of blood transfusion between the groups. Hb levels were significantly higher in the IV iron group at the time of surgery, with a greater proportion in the IV iron group achieving Hb normalization on the day of surgery compared to oral iron (90% vs. 75%; $p = 0.048$). Secondary endpoints did not show differences in the rate of complications (including infection) or postoperative length of hospital stay. A secondary manuscript suggested that compared to oral iron, IV iron improved quality of life both on the day of surgery and on the first outpatient visit (2–3 months after hospital discharge) [50].

FIT enrolled 202 patients with anemia defined as WHO definitions and iron deficiency defined as transferrin saturation (TSAT) < 20% who underwent curative resection for colorectal cancer [49]. Patients were divided into two groups receiving either 1000 mg to 2000 mg IV FCM in a single dose within four weeks of surgery or 200 mg of oral ferrous fumarate three times daily until the day before surgery and followed for 6 months. Oral iron was continued if anemia persisted after surgery. The primary endpoint reported that IV iron was not superior to oral iron in normalizing Hb levels on the day of surgery. However, IV iron led to a significant increase in Hb normalization at 1 month with 60% participants compared with 21% with oral iron (RR 2.92; 95% CI: 1.87–4.58; $p < 0.0001$), 2 months with 76% IV iron compared with 45% oral iron (RR 1.69; 95% CI: 1.29–2.23; $p < 0.0001$), and 3 months with 76% IV iron compared with 43% with oral iron (RR 1.76; 95% CI: 1.34–2.32; $p < 0.0001$). No differences were observed in RBC transfusions, postoperative complications, length of hospital stay, and mortality from baseline before surgery and postoperatively. Overall, FIT observed that IV iron increases Hb preoperatively, with the optimal benefits seen 1–3 months after surgery. These results paint an encouraging picture, with 33% of patients receiving adjuvant chemotherapy. IV iron was associated with sustained benefit three months after single-dose treatment.

### 3.2. Adjuvant Therapy after Surgery

One RCT evaluated 44 anemic women with Hb < 109 g/L and gynecological cancer who received platinum-based adjuvant chemotherapy after previous resection [32]. All patients received RBC transfusion prior to chemotherapy in accordance with the standard hospital protocol. Patients were randomly assigned to receive 200 mg of IV iron sucrose (IS) in a single dose or 200 mg of oral ferrous sulfate three times daily and followed until the next cycle of chemotherapy. The results revealed a lower requirement for RBC transfusions in the IV iron group vs. the oral iron group (22.7% vs. 63.6%, respectively, $p < 0.01$) in the next chemotherapy cycle. Significantly higher Hb and hematocrit were reported in the IV

iron group. There was no change in total quality of life scores before and after treatment in both groups.

### 3.3. Adjuvant Therapy with ESAs

Ten RCTs evaluated the effects of adding IV iron to ESAs in cancer patients with anemia related to cancer and/or chemotherapy [28–31,38–40,44,47,48]. A variety of cancers, including solid tumors (breast cancer, colorectal, lung, and gynecological) and lymphomas, were evaluated. Most studies included anemia defined as Hb < 110 g/L, and seven studies included serum ferritin and TSAT parameters, although the definitions are heterogeneous. Overall, the increase in Hb was the primary endpoint. Secondary endpoints were transfusion rate and quality of life scores. The use of IV iron was associated with significantly increased Hb levels, defined as ≥20 g/L increase from baseline and/or decreased requirement for RBC transfusion (20%) compared to oral iron and ESAs alone.

Detailed Description of Specific Trials

Auerbach et al. were the first to explore the addition of iron therapy to ESAs in the oncology setting [29]. One hundred and fifty-seven patients were randomized to four groups receiving either no iron, 325 mg of oral ferrous sulfate twice daily, 100 mg of IV iron dextran bolus injection at each visit (to the calculated dose of iron replacement), or IV iron dextran (total dose infusion calculated with a formula), and followed up for 6 weeks. All patients received 40,000 U of ESAs once weekly. All treatment groups showed a significant increase in Hb from baseline to 6 weeks. The mean increase was 9 g/L, 15 g/L, 25 g/L, and 24 g/L. The two IV iron groups showed a three-fold increase in mean change in Hb. There was a significant improvement in quality of life scores in both IV iron groups.

Henry et al. enrolled 187 patients undergoing chemotherapy and randomized in a 1:1:1 ratio to receive either 125 mg of IV sodium ferric gluconate complex (FG) weekly, 325 mg of oral ferrous sulfate three times daily, or no iron for 8 weeks [40]. All groups were scheduled to receive 40,000 U of ESAs once weekly. The primary endpoint was met with an increase in Hb without transfusions at 10 weeks. A total of 73% of the IV iron group, 46% of the oral iron group, and 41% with no iron group had a hematopoietic response defined as Hb response (increase > 20 g/L).

Bastit et al. assessed 396 patients with non-myeloid malignancies undergoing chemotherapy and Hb < 110 g/L to receive either 200 mg of IV iron + ESAs every three weeks or ESAs every three weeks and standard practice for 16 weeks [31]. A higher proportion achieved the Hb target in the IV iron group (86% vs. 73%), defined as Hb ≥ 120 g/L or Hb increase of ≥20 g/L from baseline. Treatment was tolerated in both groups, with no differences in serious adverse events being observed.

Auerbach et al. further conducted a double-blind, 2 × 2 factorial RCT in which 243 oncology patients with anemia were randomized to one of four groups (300 µg or 500 µg of ESAs once every three weeks) and (400 µg of IV iron or no iron every three weeks) and followed up for 15 weeks [30]. At the end of the intervention period, the average change in Hb was greater in the IV iron groups, and the proportion of patients who achieved Hb was higher in the IV iron groups compared to those without iron. IV iron was associated with improved quality of life scores. The median time to clinical improvement was shorter in the IV iron group than in the no IV iron group (7 weeks vs. 10 weeks).

Anthony et al. evaluated 375 oncology patients with Hb ≤ 100 g/L [28]. Patients were randomized to receive either IV iron sucrose 3 times per week with ESAs or ESAs only and followed up for 12 weeks. At the end of the intervention, Hb and QoL were significantly higher in the IV iron group compared to the no iron group.

Steensma et al. enrolled 502 patients with non-myeloid malignancies undergoing chemotherapy with Hb < 110 g/L to receive either IV FG every 3 weeks, oral iron, or oral placebo at a 1:1:1 ratio for 16 weeks [48]. The primary endpoint was the proportion of those who achieved a Hb response ≥ 20 g/L from baseline or Hb < 120 g/L in the absence of transfusions during the preceding 4 weeks. The investigators reported no difference

in Hb response, transfusion rate, or quality of life scores in the intention-to-treat cohort. Subsequent per-protocol analysis evaluating those who received at least four of five doses of planned FG doses reported that IV iron was more efficacious in increasing Hb response (80% vs. 67% vs. 65%) and decreasing transfusion rates (9% vs. 13% vs. 13%) [51].

### 3.4. Adjuvant Therapy without ESAs

Four RCTs compared the safety and efficacy of IV iron monotherapy to treat chemotherapy-induced anemia [27,39,45,46]. Patients with lymphoid, colon, solid, and non-myeloid cancers were included. Most studies defined anemia below WHO classification, and Hb increase was the primary endpoint. Hedenus et al. were the first to investigate whether IV iron would increase Hb in patients with lymphoid malignancies and functional iron deficiency (Hb 85–105 g/L and SF > 30 µg/L for women or >40 µg/L for men, TSAT $\leq$ 20%). Seventeen were randomized to receive IV iron or no treatment (standard care) and followed up at 4, 6, and 8 weeks. The study reported that IV iron significantly increased Hb at the 8-week primary endpoint. Median Hb increase was 21 g/L (range 2–35 g/L) in the IV iron group vs. 9 g/L (range 3–22 g/L). No treatment-related adverse events were reported. AnsariNejad et al. evaluated patients with stage III/IV colon cancer undergoing FOLFOX chemotherapy with Hb $\leq$ 120 g/L for women and $\leq$130 g/L for men, SF < 30 µg/L. Sixty patients were randomly assigned to receive single-dose IV iron or oral iron for 8 weeks. There was a significant increase in Hb at the primary endpoint, defined as 6 weeks after treatment with IV iron and 8 weeks of oral iron treatment (138.6 $\pm$ 7.4 g/L vs. 116.7 $\pm$ 12.8 g/L). Noronha et al. enrolled 148 patients undergoing chemotherapy with Hb < 120 g/L and at least one characteristic ID: SF < 100 µg/L, TSAT < 20%, or hypochromic RBC > 10% were enrolled. Patients were randomly assigned to receive IV iron or oral iron. The study found no difference in the mean increase in Hb at 6 weeks, and transfusion rate and quality of life scores were similar between the two arms. More recently, Makharadze et al. enrolled 244 patients who underwent chemotherapy with Hb 80–110 g/L to receive either IV iron or placebo and follow up for 18 weeks. The primary endpoint was defined as a decrease in Hb = 5 g/L from baseline at weeks 3 to 18. IV iron was associated with a maintained Hb (50.8% vs. 35.3%; *p* = 0.01). IV iron was well tolerated.

### 3.5. Radiotherapy

Two RCTs evaluated IV iron in patients with cervical cancer who underwent chemoradiotherapy or primary radiotherapy [34,42]. Kim et al. evaluated 75 women with Hb $\leq$ 120 g/L and locally advanced cervical cancer undergoing concurrent chemoradiotherapy. Patients were randomized to receive either IV iron or no iron at the start of each cycle of treatment. IV iron was administered when Hb levels = 100–120 g/L. The results revealed a significant reduction in transfusion rates and units in the IV iron arm over a six-cycle treatment period and increased Hb levels. Dreyer et al. enrolled 43 patients scheduled for primary radiotherapy with Hb $\leq$ 120 g/L and randomized to receive either IV iron + limited transfusion or standard transfusion. In the IV iron and limited transfusion group, a steady rise in Hb over 12 weeks was observed, while the transfusion and no iron group showed a 5 g/L decline in Hb after treatment.

### 3.6. Palliative

A recent study explored the feasibility of performing a large definitive RCT in patients with advanced solid tumors [33]. Study authors Dickson et al. conducted a double-blind, placebo-controlled trial in which 34 patients with anemia defined by WHO definitions with fatigue and performance status <2 were randomized in a 1:1 ratio to receive either IV iron or placebo (normal saline) and followed up for 8 weeks. The trial was feasible according to the recruitment rate (47%) and study attrition (26%). Hb increased in the IV iron group at 4 and 8 weeks. Hb levels increased in the IV group (39% vs. 8%) compared to the placebo group at 8 weeks.

## 4. Discussion

The aim of this review is to identify, describe, and organize the RCTs in IV iron therapy to treat cancer patients with anemia and outcomes. We identified 23 RCTs that evaluated IV iron therapy. Overall, IV iron increased Hb and may reduce the need for blood transfusions across oncological treatment settings, including surgery, adjuvant therapies with and without ESAs co-intervention, radiotherapy, and palliative.

In the surgical setting, all studies extant were conducted on patients with colorectal cancer who underwent primary resection surgery. IV iron increased Hb before and after surgery; however, the studies did not find a net clinical difference in postoperative outcomes, including length of hospital stay, complications, and transfusion rates. The increase in Hb levels was greater after surgery. A systematic review and meta-analysis in preoperative IV iron therapy in colorectal cancer undergoing surgical resection reported that preoperative IV iron reduces the risk of receiving RBC transfusions (Lederhuber, 2023—in press). Lederhuber et al. reported that preoperative IV iron reduced the risk of receiving RBC transfusions (RR: 0.62, 95% CI 0.41 to 0.93, $p = 0.03$, $I^2 = 0\%$, $t^2 = 0$) and increased Hb preoperatively ($g$ 0.52, 95% CI 0.08 to 0.96, $p = 0.03$, $I^2 = 66\%$, $t^2 = 0.07$). There was no significant impact on mortality, hospital length of stay, or postoperative major complications (Lederhuber, 2023—in press). One study reported that IV iron monotherapy reduced transfusion rates in patients due to commencing adjuvant chemotherapy. We did not identify any studies exploring IV iron in the neoadjuvant–surgical treatment pathway. In the recently published FIT trial, Talboom et al. reported an improvement in Hb levels in two-thirds of the IV iron group one month after surgery, and Hb levels were sustained three months after surgery [49]. One-third of the cohort went on to receive adjuvant chemotherapy. The benefit of IV iron and time to chemotherapy from surgeries requires further study.

Previous systematic reviews have summarized the evidence for the efficacy of IV iron added to ESAs [15]. In oncology patients with anemia, the goal is to improve quality of life (especially fatigue) and to complete adjuvant therapies. A positive effect on quality of life has been reported in four trials [28–30,50]. Eight trials reported no differences in quality of life outcomes. One feasibility study that evaluated IV iron therapy and placebo to manage anemia and fatigue symptoms in cancer patients was not powered to detect a change in the quality of life outcome measures [33]. To our knowledge, this is the first mapping review to summarize IV iron therapy along the cancer treatment journey. Recent studies in the surgical and palliative settings demonstrate IV iron can be administered safely throughout the cancer treatment pathway.

## 5. Conclusions

This mapping review of RCTs assessed IV iron therapy in cancer patients and outcomes. While there are different mechanisms behind anemia in cancer patients, IV iron can increase Hb and can be administered safely throughout the cancer treatment pathway from primary surgery to the palliative setting. Further studies are required to demonstrate net clinical outcomes for cancer patients.

**Author Contributions:** Conceptualization, J.L. and T.R.; methodology, J.L. and T.R.; validation, J.L., B.M. and A.A.-S.; data curation, J.L. and A.A.-S.; writing—original draft preparation, J.L. and T.R.; writing—review and editing, J.L., M.A., B.M., A.A.-S. and T.R. All authors have read and agreed to the published version of the manuscript.

**Funding:** This research received no external funding.

**Institutional Review Board Statement:** Not applicable—this article reviewed published RCTs.

**Informed Consent Statement:** Not applicable.

**Data Availability Statement:** No new data were created or analyzed in this study. Data sharing is not applicable to this article.

**Conflicts of Interest:** M.A. reports receiving funding from Covis Pharma for data management, and honoraria and meeting support from Pharmacosmos for educational and nonpromotional programs. TR reports grants from NHMRC, Vifor Pharma, and Pharmacosmos. Other authors declare no conflicts of interest.

## Appendix A

Search strategies

**The Cochrane Central Register of Controlled Trials**

#1 MeSH descriptor Iron Compounds explode all trees

#2 MeSH descriptor Ferric Compounds explode all trees

#3 MeSH descriptor Ferrous Compounds explode all trees

#4 iron OR ferrous OR ferric

#5 (#1 OR #2 OR #3 OR #4)

#6 MeSH descriptor Anemia explode all trees

#7 anemi* OR anaemi*

#8 (#6 OR #7)

#9 (#5 AND #8)

**MEDLINE (PubMed)**

("Iron Compounds"[Mesh] OR "Ferric Compounds"[Mesh] OR "Ferrous Compounds"[Mesh] OR iron OR ferrous OR ferric) AND ("Anemia"[Mesh] OR anemi* OR anaemi*) AND ((randomized controlled trial [pt] OR controlled clinical trial [pt] OR randomized [tiab] OR placebo [tiab] OR drug therapy [sh] OR randomly [tiab] OR trial [tiab] OR groups [tiab]) NOT (animals [mh] NOT humans [mh]))

**EMBASE (Ovid SP)**

1. exp iron therapy/
2. (iron or ferrous or ferric).af.
3. 1 or 2
4. exp anemia/
5. (anemi* OR anaemi*).af.
6. 4 or 5
7. exp crossover-procedure/or exp double-blind procedure/or exp randomized controlled trial/or single-blind procedure/
8. (random* or factorial* or crossover* or placebo*).af.
9. 7 or 8
10. 3 and 6 and 9

**ISI Web of Science: Science Citation Index-Expanded (SCI-EXPANDED) and Conference Proceedings Citation Index-Science (CPCI-S)**

# 1 TS = (iron OR ferrous OR ferric)

# 2 TS = (anemi* OR anaemi*)

# 3 TS = (random* OR rct* OR crossover OR masked OR blind* OR placebo* OR metaanalysis OR systematic review* OR meta-analys*) # 4 #3 AND #2 AND #1

**CINAHL Plus (EBSCO)**

S1 (MH "Clinical Trials+")

S2 PT Clinical trial

S3 TX clinic* n1 trial* or TX ((trebl* n1 blind*) or (trebl* n1 mask*)) or TX ((tripl* n1 blind*) or (tripl* n1 mask*))

S4 TX ((singl* n1 blind*) or (singl* n1 mask*)) or TX ((doubl* n1 blind*) or (doubl* n1 mask*))

S5 TX randomi* control* trial*

S6 (MH "Random Assignment")

S7 TX random* allocat*

S8 TX placebo*

S9 (MH "Placebos")

S10 (MH "Quantitative Studies")

S11 TX allocat* random*
S12 S1 or S2 or S3 or S4 or S5 or S6 or S7 or S8 or S9 or S10 or S11 S13 (MH "Iron")
S14 (iron OR ferrous OR ferric)
S15 S13 or S14
S16 (MH "Anemia+")
S17 (anemi* OR anaemi*)
S18 S16 or 17
S19 S15 and S18
S20 S12 and S20
**Trial registries**
(anemi* OR anaemi*) AND (iron OR ferrous OR ferric)

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
