# Peer review of "Intravenous Iron Therapy to Treat Anemia in Oncology: A Mapping Review of Randomized Controlled Trials"

_curroncol, doi:10.3390/curroncol30090569_

Round 1
Reviewer 1 Report
Please, see below.
In their work, Lim J. et al. report a systematic review of RCTs that have explored the effects of IV iron replacement therapy in patients with cancer-related anemia. The studies were grouped according to the type of anti-cancer treatment received by the patients (surgery only, surgery + adjuvant therapy, RT, palliative care). The systematic review deals with a condition of clinical relevance. The topic of the manuscript is addressed entirely and methodologically accurately. The bibliography is up-to-date and relevant.
My comments are:
· The manuscript's organization (a division of RCTs according to treatment) is valuable and informative for clinicians. In this regard, I ask the authors if they could consider adding to Figure 1, next to each treatment tile, the number of the references who have explored iron treatment in that specific treatment setting.
· It is evident that the authors have an excellent knowledge of the topic! However, some parts of the manuscript may be more challenging for a less experienced reader, who may not know the studies summarized in the review in such detail. Therefore, I suggest better explicit the treatments received by patients (type of IV/oral iron, ESAs, duration of treatment, etc.), limit the acronyms as much as possible, and standardize the units of measurement (e.g., for Hb) indicated in the manuscript. Here are some examples in detail:
- Line 66 and 76: I would uniform the description of the comparators (I prefer comparators as reported in line 76)
- Line 68 “which were included in the previous meta-analysis”: please add ref.
- Line 74: Hb is reported in g/dl, while previously Hb level was reported in g/l (line 27): please, standardize the units of measurement across the whole manuscript
- Line 150 “RBC” appears for the first time in the manuscript; I would specify this acronym at the beginning of the manuscript; for example, in line 44, you could write “red blood cell (RBC) transfusion”
- In Line 141 “TSAT” and in Line 222 “SF” are used for the first time: please specify “transferrin saturation” and “serum ferritin”
- Line 228: remove )
- Line 233: please, authors correct units of serum ferritin (mg/l instead of mcg/ml)
- Line 244: 75… subjects?
Concerning treatment, Line 143 reports “three 200 mg of oral iron”; Line 179 reports “IV iron bolus or IV iron total dose infusion”; Line 196 reports “ESA 300μg Q3W 196 or ESA 500 μg Q3W”; I ask the authors to explain the treatment (dose, timing, types) received by patients. In general, could authors specify the type of IV/oral iron and ESAs used in different studies? Line 214: ferric gluconate is the only specified preparation across the manuscript…
Concerning the paragraphs; I think they could be title them as follow:
- 3.1 IV iron in surgery
- 3.2 IV iron monotherapy in patients treated with adjuvant ChT after surgery
- 3.3 IV iron plus ESAs in patients treated with adjuvant ChT after surgery
- 3.4 is the same as 3.2?
- 3.5 IV iron in radiotherapy
- 3.6 IV iron in the palliative care setting
Moreover, I would remove the subdivision of paragraphs 3.1.1 and 3.3.1.
Other minor points:
Line 31: “Reduced levels of Hb” could be better expressed as “severity of anemia”
Line 45 and 48: please uniform the use of ESA or ESAs (across the whole manuscript)
Line 125 “correct laboratory values of anaemia”: I think it could be better to say “correct laboratory values of Hb”
Bibliography: I suggest the authors consider whether they deem it appropriate to mention “Management of anaemia and iron deficiency in patients with cancer: ESMO Clinical Practice Guidelines” by Aapro et al., 2018 in the Introduction
Reviewer 2 Report
The manuscript is well written where the authors performed systematic mapping review of RCTs to evaluate IV iron therapy in patients with cancer and anemia. However, the authors need to address the following comments.
1. The dosages in supplementary table 1 varies in different studies. If there is any specific dosage that can be identified based on those studies to administer to the patients.
2. The authors need to include data if there is any adverse effect related to IV iron delivery in the studied patients.
Reviewer 3 Report
This is a well-written systematic mapping review of randomized controlled trials in the iron therapy/oncology sphere.
Studies were selected appropriately, the process by which the reports to be analyzed in detail were identify is laid out clearly. Discussion of studies is appropriately supported by the evidence, as are the conclusions reported by the authors.
Issues observed are minor and mostly matters of clarification.
1. In the Abstract line 9, it would probably be more accurate to say "psychological" rather than "mental".
2. In the Abstract wind 13, it should say "median number of patients…".
3. In Introduction, the sentence beginning “The cause of anemia…” in line 27 and following, is confusing and needs to be rewritten. While anemia in cancer patients is certainly multifactorial, direct causes would include blood loss in gastrointestinal tumors and bone marrow replacement by cancer cells. Bone marrow suppression, cancer -related cachexia, and hepcidin-mediated iron sequestration are all indirect factors mediated by inflammatory cytokines induced in response to cancer.
4. In line 39, the words, “most…” should be followed by “… patients being…”.
5. In line 54, rather than referring to “the old iron dextran” it would be more accurate to refer to “high molecular weight iron dextran products no longer available for clinical use” or words to that effect.
6. In paragraph 3.1, the authors should explicitly state at the beginning that all five studies reviewed involved colorectal cancer. As it is written, the reader might assume that the fifth study discussed did not necessarily involve colorectal cancer.
7. The title for section 3.1.1 should be changed to something like “Detailed description of IVICA and FIT trials”
8. In the same section, the citations for the studies referred to above should be inserted into the discussion of these studies.
9. In section 3.2, the quantity of the single-dose IV iron should be stated (line 160).
10. In section 3.3.1, the title should be changed to something like “Description of specific studies”.
11. In Discussion, the sentence beginning “IV iron…” (line 269) should be modified to say that “the studies did not find a clinical difference in post operative outcomes apart from the degree of anemia…”.
There are some minor issues of English usage.
1. Usage of "anemia" vs. "anaemia" is not consistent.
2. There are several instances where incorrect verb tenses are employed. ("Has" vs. "Have"; "Are" vs. "Were")
3. There are some instances where commas are inserted in the wrong places.
Reviewer 4 Report
Abstract:
The abstract provides a clear overview of the significance of the problem (anemia in cancer patients) and its most common cause (iron deficiency).
It also gives a brief understanding of the purpose of the review (systematic mapping review of RCTs regarding intravenous iron therapy for anemia in cancer patients) and the overall findings.
The exact number of RCTs, median number of participants, and IQR have been mentioned, which is good.
The distribution of the RCTs based on different treatment stages or categories (surgical outcomes, adjuvant therapies, radiotherapy, and palliative care) is delineated, offering the reader a quick understanding of the spread of the studies.
Recommendations:
•It might be beneficial to state why newer intravenous iron formulations are considered more favorable, even briefly.
•"More studies are needed to demonstrate net clinical outcomes." – This line could be expanded upon to hint at what 'net clinical outcomes' encompass or what the existing gaps might be.
Introduction
The introduction provides a broad overview of the problem of anemia in cancer patients, its causes, implications, and the consequences on treatment, especially surgery.
•It also sheds light on the current treatment options for anemia and highlights the shift towards IV iron therapy due to the shortcomings of other treatments.
Recommendations:
•Although the introduction does well in setting the scene, it might be beneficial to briefly mention why IV iron therapy, specifically in cancer patients (as opposed to other patients suffering from anemia) is of interest or significance. This would further enhance the rationale for the study.
•There's a lot of focus on the preoperative phase, and while that's crucial, ensuring that other phases are similarly detailed might provide a broader picture.
Methods
Research Design:
•The authors have followed the PRISMA statement for systematic reviews and meta-analyses,. Their registration with PROSPERO also demonstrates a commitment to transparency and credibility. I am concerned about this part as it seems this is a scoping and not a systematic review using PRISMA Extension for Scoping Reviews (PRISM should be usedA-ScR).
Eligibility Criteria:
•The inclusion and exclusion criteria cover population, intervention, comparators, and outcomes (PICO). However, as it seems to be a scoping review, it should follow the PCC.
Search Strategy:
•A wide range of databases is utilized, demonstrating a rigorous and extensive search approach. Not limiting by language or publication status can help capture all relevant articles.
Study Selection and Data Extraction:
•Using two independent reviewers for study selection is a solid method to reduce potential bias. The structured approach of initial screening and then full-text assessment ensures methodical evaluation of articles. Including software tools like Rayyan and EndNote provides a structured and standardized method for data extraction and referencing.
Data Synthesis:
•The decision to present results in a summary table and through a narrative synthesis is appropriate.
Major Concern:
•Upon reflection, the approach appears more consistent with what one might expect from a scoping review rather than a systematic mapping review. The distinction is significant. The methodology seems to lack alignment with specific guidelines such as those provided by JBI (Joanna Briggs Institute) or Cochrane, which offer structured pathways for conducting reviews. Given this, there's a bit of ambiguity about the authors' chosen path, which makes it challenging to assess the rigor of the review process fully. For example, as a systematic review, the authors should perform a quality assessment - Critically appraise included studies. But in reality, authors only what to map the existence of evidence, which is more adjustable to a scoping review.
It is critical for the authors to clearly define their review type and ensure alignment with the chosen methodology's key characteristics. The reasons and justifications should be explicit to maintain clarity and reproducibility if it's a hybrid or modified approach.
Recommendations:
• Consider providing a more explicit rationale or justification for the chosen review methodology. If deviations from standard guidelines occur, these should be explicitly stated and justified.
•The data synthesis section might benefit from more details about handling and analysing data. Were there specific metrics or themes that were pivotal in the analysis?
•If the work is more aligned with a scoping review, reframing the review in that context or providing clear justifications for the systematic mapping approach might be beneficial.
The results presented in the manuscript are generally clear and well-organized. However, some areas could be improved to enhance the clarity and impact of the findings.
In the discussion section, the authors have appropriately summarized the review's key findings and provided insights into the use of IV iron therapy in various oncological treatment settings. The discussion addresses the efficacy of IV iron in increasing hemoglobin levels and potentially reducing the need for blood transfusions across different cancer treatment stages. The section also highlights the lack of net clinical differences in specific postoperative outcomes and the potential benefits of IV iron therapy in improving quality of life in some studies.
However, some areas could be further improved in the discussion:
1. Although the authors briefly mention that a systematic review and meta-analysis on preoperative IV iron therapy in colorectal cancer surgery reported reduced risk of receiving RBC transfusions, they could consider providing more details.
2. The discussion on the recently published FIT trial (Talboom et al.) could be expanded to include more specific details about the study design, patient population, and critical findings. This will help readers better understand the relevance of this trial in the context of IV iron therapy.
3. The authors mention that one-third of the cohort from the FIT trial went on to receive adjuvant chemotherapy. Still, there is no further discussion on the potential implications of this finding or how it may impact the use of IV iron therapy in this patient group. Further exploration of this aspect could strengthen the discussion.
Regarding the conclusions, the authors have summarized the review's main findings concisely. The conclusion correctly emphasizes the potential benefits of IV iron in increasing hemoglobin levels and its safe administration throughout the cancer treatment journey. However, it could be strengthened by explicitly mentioning that further studies are needed to demonstrate the net clinical outcomes of IV iron therapy in cancer patients.
Overall, the discussion and conclusions largely align with the results presented in the review. By addressing the above points and potentially expanding on certain aspects, the authors can further enhance the clarity and impact of the discussion and conclusion sections in the manuscript.
As mentioned before, my primary concern is about the method. It seems, to me, a Scoping review since the authors want to map the existing literature, and no assessment of the quality of the included studies was performed.
Author Response
Please see the attachement

Round 2
Reviewer 1 Report
Thanks to the authors for submitting the requested changes. In my opinion, the manuscript currently appears more understandable and fine for publication.
Reviewer 2 Report
The authors have answered the questions.
Reviewer 3 Report
The authors have been highly responsive to reviewer comments.